# Clinical Significance of Intraoperative Exposure of Inferior Alveolar Nerve during Surgical Extraction of the Mandibular Third Molar in Nerve Injury

**DOI:** 10.3390/jcm10194379

**Published:** 2021-09-25

**Authors:** Sung-Woon On, Seoung-Won Cho, Soo-Hwan Byun, Byoung-Eun Yang

**Affiliations:** 1Division of Oral and Maxillofacial Surgery, Department of Dentistry, Hallym University Dongtan Sacred Heart Hospital, Hwaseong 18450, Korea; drummer0908@hanmail.net; 2Graduate School of Clinical Dentistry, Hallym University, Chuncheon 24252, Korea; kotneicho@gmail.com (S.-W.C.); purheit@daum.net (S.-H.B.); 3Institute of Clinical Dentistry, Hallym University, Chuncheon 24252, Korea; 4Division of Oral and Maxillofacial Surgery, Hallym University Sacred Heart Hospital, Anyang 14066, Korea

**Keywords:** third molar surgery, cone-beam computed tomography, peripheral nerve injury, inferior alveolar nerve, oral surgery

## Abstract

During extraction surgery, the inferior alveolar nerve (IAN) can occasionally be observed in the extraction socket of the mandibular third molar (M3). The purpose of this study was to investigate and compare the incidence of IAN injury in groups with and without intraoperative IAN exposure during surgical extraction of M3, and to identify additional risk factors for the IAN injury in addition to the IAN exposure. A total of 288 cases in 240 patients, who underwent surgical extraction of M3 by a single surgeon, were divided into the exposed group (*n* = 69) and the unexposed group (*n* = 219). The surgeon recorded the information regarding the procedure when the clinical observation of IAN exposure was made during the surgery. The incidence of IAN injury after the extraction surgery was significantly higher in the exposed group than in the unexposed group (4.3% versus 0%, *p* < 0.05). Paresthesia was recognized in three cases of the exposed group, but it showed complete recovery within three postoperative months. No case of permanent paresthesia was detected in both groups. According to the logistic regression, the only significant risk factor of IAN injury in the exposed group was the increase of age (OR 1.108, *p* < 0.05). Intraoperative IAN exposure during surgical extraction of M3 may show a higher incidence of IAN injury than the case without IAN exposure, representing an incidence of 4.3%. Even if the paresthesia associated with IAN exposure occurs, it is likely to be a temporary injury, and this risk may increase with age.

## 1. Introduction

Surgical extraction of the mandibular third molar is prevalent in oral and maxillofacial surgery, but some complications may occur. The most significant complication is postoperative nerve injury, associated with inferior alveolar nerve (IAN) or lingual nerve after the surgical extraction of the mandibular third molar. According to the literature, the incidence of IAN damage and lingual nerve damage after third molar surgery have been reported to be various, at approximately 0.35–8.4% [1,2,3] and 0.02–2% [4]. Consequently, several surgical approaches have been proposed to reconstruct the damaged tissues, with promising short-term outcomes [5].

In some cases, IAN can be directly observed in or around the extraction socket of the mandibular third molar during the extraction. The incidence of IAN exposure during extraction of the mandibular third molar has been reported to be 5.7–43% [1,6], and the resulting temporary paresthesia and permanent paresthesia have been reported to be 14.3–22% [6,7,8,9] and less than 1% [1,10,11,12], respectively. As described above, several studies have reported the frequency of IAN exposure and the rate of related nerve damage after extraction of the mandibular third molar. However, no study has directly compared cases with and without intraoperative IAN exposure during surgical extraction of mandibular third molar to determine how nerve exposure is related to nerve damage. Therefore, it is necessary to demonstrate whether IAN exposure has clinical significance in nerve injury through this method. In addition, when IAN injury occurs, it is necessary to investigate whether there are additional risk factors besides nerve exposure.

The aims of the present study were: (1) to investigate and compare the incidence of IAN injury in groups with and without intraoperative IAN exposure during surgical extraction of the mandibular third molar; (2) to identify additional risk factors for the IAN injury in addition to the IAN exposure.

## 2. Materials and Methods

### 2.1. Samples and Group Assignment

This retrospective study was conducted on patients who underwent mandibular third molar extraction from January 2019 to April 2021 by a single surgeon (S.W.O) in Hallym University Dongtan Sacred Heart Hospital. A total of 795 patients with 885 mandibular third molars underwent the mandibular third molar extraction during the study period. The inclusion and exclusion criteria are as follows. 

Inclusion criteria:(1)Patients who were medically healthy and over 16 years of age(2)Patients who underwent the surgical extraction of mandibular third molar under local anesthesia(3)Patients who visited the hospital for suture removal and follow-up check(4)Patients whose preoperative panoramic view was available and whose preoperative cone-beam computed tomography (CBCT) was available only when the mandibular third molar and the IAN canal overlapped on the panoramic radiograph

Exclusion criteria:(1)Patients who underwent the simple extraction of mandibular third molar or surgical extraction of mandibular third molar under general anesthesia(2)Patients whose adjacent molars were extracted simultaneously(3)Patients with mandibular third molar associated with periapical or cystic lesions or tumors(4)Patients taking steroids for medical problems

Among them, 404 cases were excluded, and the remaining 481 cases were analyzed radiologically. After the radiological analysis using preoperative panorama and CBCT, cases in which the IAN was not exposed during surgery and the mandibular third molar contacted the IAN were additionally excluded. Consequently, a total of 288 cases in 240 patients were finally included in this study. The detailed process of case selection is presented in Figure 1.

This retrospective study was reviewed and approved by the Institutional Review Board of Hallym University Dongtan Sacred Heart Hospital, Hwaseong, Korea (IRB No. 2021-04-012).

The samples were divided into two groups: the exposed group (n = 69; mean age, 29.32 years; Table 1) and the unexposed group (n = 219; mean age, 29.25 years; Table 1). The cases where IAN was observed during surgical extraction of the mandibular third molar were classified as the exposed group. In all cases of the exposed group, contact between the IAN and the mandibular third molar was observed radiologically. The radiological definition of “contact” was the absence of an interstitial space or bone marrow between the mandibular third molar and the corticated or uncorticated IAN canal as suggested by Ohman et al. [13]. The cases in which the IAN was not exposed during the operation and the mandibular third molar did not contact the IAN radiologically were classified into the unexposed group.

### 2.2. CBCT Image Acquisition and Assessment of the Relationship between the IAN Canal and Mandibular Third Molar in the Exposed Group

CBCT scans were taken preoperatively using the Alphard VEGA dental CT system (Asahi Roentgen Ind. Co., Ltd., Kyoto, Japan) in 80 kV and 5 mA (Scanning time, 18 s; field of view, 154 mm × 154 mm; slice, 0.3 mm) when the preoperative panoramic radiograph showed signs of proximity between the IAN canal and the mandibular third molar. Three-dimensional image reconstruction was performed through OnDemand 3D (Cybermed Inc., Seoul, Korea) for the assessment of the relationship between the IAN and mandibular third molar on the coronal view. The course of the IAN canal (Figure 2) and the relationship of the mandibular third molar to the lingual cortex (Figure 3) were investigated in the exposed group based on the criteria suggested by Ohman et al. [13] to evaluate the radiological properties and identify additional factors for nerve injury.

### 2.3. Surgical Procedure and Identification of IAN Exposure

Surgical extraction of the mandibular third molar was performed by a single surgeon. All patients underwent the procedure under local anesthesia (lidocaine 2% with 1:100,000 epinephrine), and a mucoperiosteal incision and buccal flap reflection were carried out. Bone reduction around the tooth and odontomy (tooth sectioning) using rotary instruments were performed when necessary. Immediately before the surgery, if preoperative CBCT was present, the CBCT image was reviewed with the patient. When the IAN and the mandibular third molar were in contact in CBCT images, the surgeon inspected all extraction socket walls and floors to identify the IAN exposure after copious irrigation of the socket following the tooth removal. If the IAN bundle was observed, it was palpated using instruments (e.g., suction tip). When the patient felt abnormal sensations, including a cold sensation or pain, the clinical diagnosis of IAN exposure was made (Figure 4). The flap was then sutured, and the patients received IV antibiotics (amoxicillin/clavulanate 1.2 g or cefotetan 1 g), analgesics (ketorolac tromethamine 30 mg), and steroids (dexamethasone 5 mg) immediately after surgery, and were prescribed oral antibiotics (usually oral amoxicillin/clavulanate 375 mg 3 times daily or cefdinir 100 mg 3 times daily) and analgesics (zaltoprofen 80 mg 3 times daily) for about one week and steroids (prednisolone 5 mg 3 times daily) for two days. One week after the surgery, the sutures were removed. If the patients felt hypoesthesia or the abnormal sensation persisting even after the next day after surgery, they were asked to visit the hospital immediately. In such cases, hypoesthesia or paresthesia of the lower lip and chin were clinically investigated. When nerve injury was diagnosed, oral steroids (prednisolone 5 mg 3 times daily) were prescribed for one week and continued prescription of the vitamin B complex (cyanocobalamin 1 μg, nicotinamide 25 mg, pyridoxine hydrochloride 1 mg, riboflavin 6 mg, and thiamine hydrochloride 6 mg 3 times daily) was performed during the follow-up period. The follow-up was conducted up to 6 months from the day the paresthesia was recognized.

### 2.4. Statistical Analysis

Descriptive statistics were performed to calculate the mean and standard deviation of the variables obtained from the patient data. The Shapiro-Wilk test and Kolmogorov-Smirnov test were used to evaluate the normality of the data. Wilcoxon-signed rank test, Chi-square test, and Fisher’s exact test were used to compare differences and ratios of variables between the two groups. The influence of demographic or procedure-related variables on the occurrence of IAN injury was analyzed using univariate and stepwise multiple logistic regression. *p* values of less than 0.05 were considered statistically significant. All statistical analyses were conducted using SPSS 21.0 (SPSS Inc, Chicago, IL, USA).

## 3. Results

### 3.1. Comparison of Demographic or Procedure-Related Variables and the Incidence of IAN Injury between the Two Groups

Among the demographic or procedure-related variables, only whether the bone reduction was performed showed a significant difference between the two groups (*p* < 0.05; Table 1). There were no significant differences between the two groups in variables such as age, sex, side of tooth presence, and whether or not the odontomy was performed.

Regarding the incidence of IAN injury, the incidence was significantly higher in the exposed group than in the unexposed group (4.3% versus 0%, *p* < 0.05; Table 1). Paresthesia was recognized in three cases of the exposed group, but it was temporary paresthesia in which the sensation recovered within three months after surgery. In no case was a sign of permanent paresthesia detected.

### 3.2. Radiographic Characteristics in the Exposed Group

In the course of the IAN canal in CBCT images, the most common radiographic sign was lingual (59.4%), followed by inferior (26.1%), inter-radicular (11.6%), and buccal (2.9%) (Table 2).

Regarding the relationship of the mandibular third molar to the lingual cortex, all cases in the exposed group showed the contact between the mandibular third molar and the lingual cortex, and the frequency of contact patterns was highest in the order of thinning (62.3%) and perforation (37.7%) (Table 2).

### 3.3. Factors Affecting the Occurrence of IAN Injury after Surgical Extraction of Mandibular Third Molar in the Exposed Group

The univariate logistic regression and multiple stepwise logistic regression of the occurrence of IAN injury versus demographic or procedure-related variables in the exposed group are shown in Table 3 and Table 4, respectively. The incidence of IAN injury increased significantly only with age (OR 1.108, 95% CI 1.001–1.228, *p* < 0.05; Table 3) in the univariate logistic regression. In addition, multiple stepwise logistic regression selected age (OR 1.108, 95% CI 1.001–1.228, *p* < 0.05; Table 4) as a solitary risk factor.

## 4. Discussion

This retrospective case-control study investigated the incidence of IAN injury following the exposure or non-exposure of IAN during surgical extraction of the mandibular third molar. As a result, the exposed group exhibited a significantly higher incidence of IAN injury than the unexposed group (4.3% versus 0%, *p* < 0.05; Table 1). In particular, there was no case of IAN injury in the unexposed group. Although some cases in the exposed group showed signs of temporary IAN injury, in no case was the permanent paresthesia observed. In addition, when logistic regression analysis was performed to identify risk factors for the IAN injury in the exposed group, the risk of IAN injury increased with age (OR 1.108, 95% CI 1.001–1.228, *p* < 0.05; Table 3 and Table 4). Therefore, if the IAN is exposed during surgical extraction of the mandibular third molar, it is necessary to inform the patients that the incidence of IAN injury is higher when exposed and that the possibility of IAN injury increases with age. However, it should also be explained that continuous follow-up checks and medication is required because it is most likely to be a temporary IAN injury.

Several studies have reported IAN injury related to IAN exposure after the mandibular third molar extraction [6,7,8,9]. Tay and Go [9] reported that 20.3% of cases in which IAN was observed during surgical removal of the mandibular third molar showed paresthesia, and 71% of cases showed a recovery one year after the surgery. Tantanapornkul et al. [6] reported postoperative paresthesia in 6 out of 27 cases after the mandibular third molar extraction, resulting in an IAN injury rate of 22%. Maekawa et al. [8] conducted a study to evaluate the relationship between the mandibular third molar and IAN using CT. As a result, postoperative dysesthesia on the lower lip was observed in about 14.3% of cases exposed to IAN. Hasegawa et al. [7] also presented a 20% IAN injury in cases of IAN exposure in a study to identify risk factors for hypoesthesia of the lower lip after extraction of the mandibular third molar. In the present study, exposure-related IAN injury rate was low at 4.3% compared to other studies. This finding was an unexpected result of the authors, and the following can be considered possible reasons. First, all patients in this study received IV steroids immediately after surgery and oral steroids for two days after surgery. The early application of steroids for nerve injury is widely accepted because of their anti-inflammatory and neurotrophic effects [14,15,16,17]. Unless it is a direct nerve injury such as axonotmesis or neurotmesis by a surgical bur, injury to IAN may occur indirectly due to compression of the tissue surrounding the nerve trunk of IAN canal. Therefore, It is plausible that the administration of steroids immediately after surgery and two days after surgery in this study masked the actual IAN damage related to IAN exposure. Second, procedural factors related to the surgeon’s experience may have had an influence. It is well known that the less skilled surgeons have a significantly higher incidence of postoperative complications, including nerve paresthesia after mandibular third molar extraction [18]. Since there is a difference in the experience or skill of the operator between this study and the others, the difference in the IAN injury rate contributed by the operator factor cannot be excluded. In other studies, although surgeons with sufficient experience operated, most of them included the results of surgery performed by several surgeons, including resident doctors. In the present study, all surgeries were performed by a single staff-grade oral and maxillofacial surgeon with more than eight years of experience. Therefore, the difference between the results of this study and that of others may have been caused by the difference in the composition of the operator group, not merely the difference in the operator’s experience.

Regarding radiographic characteristics in the exposed group, the most common course of IAN canal in CBCT images was lingual (59.4%, Table 2). All cases in the exposed group showed the contact between the mandibular third molar and the lingual cortex, and thinning (62.3%, Table 2) was the most frequently observed finding of the contact patterns. Maegawa et al. [8] reported that among the seven cases with IAN exposure, three cases showed lingual (42.9%) in the buccolingual relationship between IAN and mandibular third molar and accounted for the largest proportion. Although this study also showed a result consistent with that of Maegawa et al. [8], the result of this study was derived from a larger number of samples. Most of the studies on IAN exposure related to the mandibular third molar extraction focused on the accuracy and characteristic findings on the panoramic radiograph, which was confirmed with CBCT images. To the best of our knowledge, there is no study that performed CBCT analysis on the positional relationship between IAN and mandibular third molar when IAN exposure was observed for as many samples as in this study. According to the result of the present study, it is suggested that the possibility of IAN exposure may be high when IAN and the mandibular third molar are in contact, and the IAN is located on the lingual side of the mandibular third molar on the CBCT images. Therefore, when the contact between the IAN and the mandibular third molar is suspected on the panoramic radiograph, preoperative CBCT taking and review are considered essential.

One of the unexpected results in this study was that no IAN injury occurred in the unexposed group. This is thought to be because the possibility of contact between IAN and the mandibular third molar was excluded in patients in the unexposed group when group assignment was made. Therefore, if the surgeon confirms that the IAN does not contact the mandibular third molar on the radiographic image and does not perform excessive osteotomy or aggressive odontectomy during surgery, it is possible to prevent damage to the IAN after surgical extraction of the mandibular third molar. Another unexpected result of this study was that permanent IAN damage did not occur in the exposed group. Although three patients in the exposed group showed paresthesia on the IAN-innervated area, they all recovered from their paresthesia within three months. As described above, since IAN injury associated with the extraction of the mandibular third molar is often due to compression caused by a tooth, blood clot, or tissue edema, it is presumed to be only a temporary injury even when the IAN is exposed if rotary instruments caused no direct nerve injury. Therefore, although the third molar surgery is one of the most performed in the oral and maxillofacial area, careful manipulation and techniques based on the standard surgical principle should be required. In addition, when paresthesia due to IAN injury is recognized, medications including steroids and vitamin B complex and continuous follow-up are considered necessary.

The present study is different from previous studies in that it was the first study to directly compare the incidence of IAN injury in the IAN-exposed and unexposed groups. To exclude the possibility that the operator could not find the exposed IAN in the unexposed group, the unexposed group in this study included only cases in which the contact between IAN and the mandibular third molar was not observed in the radiological images. Furthermore, the present study performed logistic regression analysis to identify additional risk factors for IAN injury only within the exposed group and thus is different from other studies in which logistic regression analysis was performed on all enrolled patients.

The present study demonstrates that when IAN is exposed during surgery, the incidence of IAN injury is low and is likely to be temporary damage. However, our study has several limitations. First, there are limitations such as the retrospective nature, small sample size, and difference in the number of cases between the two groups. Further studies, such as prospective or randomized controlled studies, are needed to determine the actual incidence of IAN exposure and the associated incidence of IAN injury when the IAN and the mandibular third molar are in contact on radiological images. The authors are planning additional studies to achieve these objectives. Second, the clinical inspection may not be the most accurate method of assessing nerve exposure. However, some researchers confirmed the IAN exposure by clinical inspection [7,9,10,19,20,21,22]. Further studies on objective quantification methods for nerve exposure are needed.

In conclusion, intraoperative IAN exposure during surgical extraction of the mandibular third molar might show a higher incidence of IAN injury than the case without IAN exposure, representing an incidence of 4.3%. Even when the paresthesia associated with IAN exposure occurs, it is likely to be a temporary injury. Additionally, the risk of paresthesia may increase with age. It is necessary to inform the patients of the clinical significance of IAN exposure during the surgery. It should also be emphasized that continuous follow-up checks and medication are required if the paresthesia occurs.

## Figures and Tables

**Figure 1 jcm-10-04379-f001:**
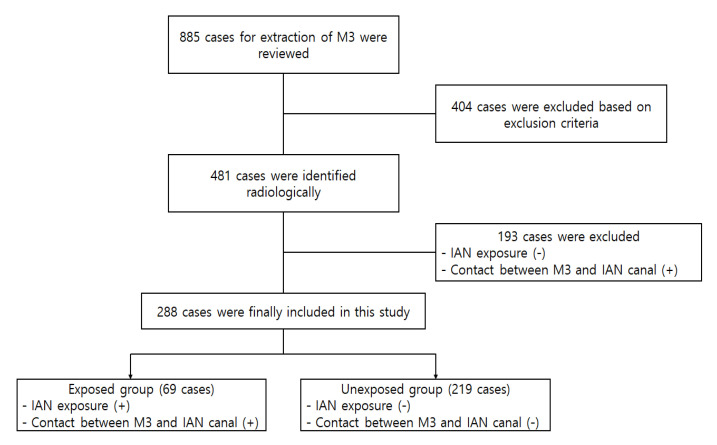
Flowchart showing the process of case selection. M3, mandibular third molar; IAN, inferior alveolar nerve.

**Figure 2 jcm-10-04379-f002:**
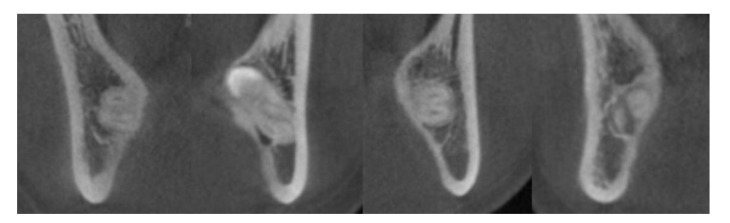
Classification of the course of IAN canal by Ohman et al. [13]: buccal (leftmost); lingual (middle left); inferior (middle right); inter-radicular (rightmost).

**Figure 3 jcm-10-04379-f003:**
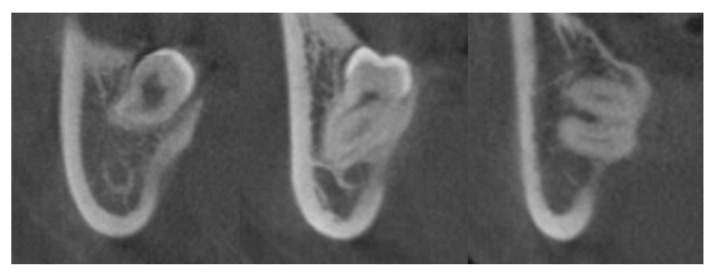
Classification of the relationship of the mandibular third molar to the lingual cortex by Ohman et al. [13]: no contact (left); contact, thinning (middle); contact, perforation (right).

**Figure 4 jcm-10-04379-f004:**
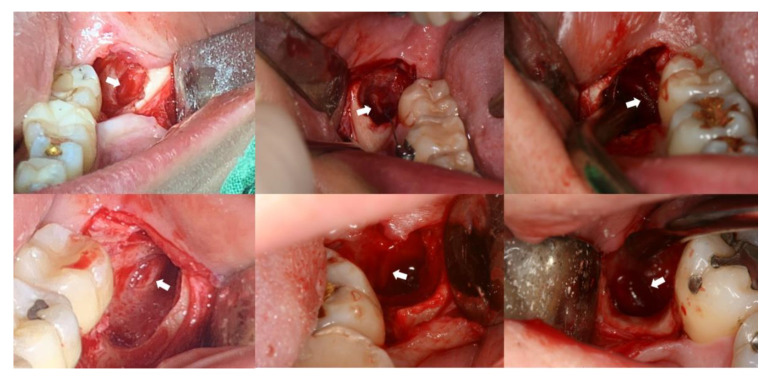
Clinical photos showing intraoperative IAN exposure (white arrows) in the extraction socket of the mandibular third molar during surgery.

**Table 1 jcm-10-04379-t001:** Comparison of Baseline Characteristics and Incidence of Inferior Alveolar Nerve Injury between the Exposed Group and the Unexposed Group.

Variables	Exposed Group	Unexposed Group	*p*-Value
	(*n* = 69)	(*n* = 219)	
Age (years)	29.32 ± 8.75 (range: 17-61)	29.25 ± 10.70 (range: 17-68)	0.393
Sex (male: female)	30:34	94:82	0.370
Side (right: left)	43:26	112:107	0.104
Procedural factors			
Bone reduction (*n*)	68 (98.6%)	186 (84.9%)	0.002 *
Odontomy (*n*)	59 (85.5%)	174 (79.5%)	0.264
Incidence of inferior alveolar nerve injury (*n*)	3 (4.3%)	0 (0%)	0.013 ^†^

* *p*-Value examined using Chi-square test; ^†^, *p*-Value examined using Fisher’s exact test.

**Table 2 jcm-10-04379-t002:** Characteristics of radiographic signs in the exposed group.

Variables	Number of Samples	%
The course of the inferior alveolar nerve canal *		
Buccal	2	2.9
Lingual	41	59.4
Inferior	18	26.1
Inter-radicular	8	11.6
The proximity of the mandibular third molar to the lingual cortex *		
No contact	0	0
Contact, thinning	43	62.3
Contact, perforation	26	37.7

* Classification suggested by Ohman et al.

**Table 3 jcm-10-04379-t003:** Univariate Logistic Regression of the Occurrence of Inferior Alveolar Nerve Injury versus Demographic and Procedural Variables in Exposed Group.

Variables	OR	95% CI	*p*-Value
Age (year)	1.108	1.001–1.228	0.048 *
Sex			
Male	Ref.		
Female	N/E		N/E
Side			
Right	Ref.		
Left	3.500	0.301–40.652	0.317
The course of the inferior alveolar nerve canal			
Buccal	Ref.		
Lingual	N/E		N/E
Inferior	1.000		1.000
Inter-radicular	N/E		N/E
The proximity of the mandibular third molar to the lingual cortex			
Contact, thinning	Ref.		
Contact, perforation	3.500	0.301–40.652	0.317
Bone reduction	N/E		N/E
Odontomy	N/E		N/E

OR, odds ratio; CI, confidence interval; N/E, not estimable because of no patient in the reference variable; ^*^, *p* < 0.05.

**Table 4 jcm-10-04379-t004:** Multivariate Logistic Regression of the Occurrence of Inferior Alveolar Nerve Injury versus Demographic and Procedural Variables in the Exposed Group.

Variables	OR	95% CI	*p*-Value
Age (year)	1.108	1.001–1.228	0.048 *
Side (left)			0.396
The proximity of the mandibular third molar to the lingual cortex (contact, perforation)			0.734

OR, odds ratio; CI, confidence interval; *, *p* < 0.05.

## Data Availability

The data that support the findings of this study are available from the corresponding author upon reasonable request.

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
