# Peer review of "Clinical Significance of Intraoperative Exposure of Inferior Alveolar Nerve during Surgical Extraction of the Mandibular Third Molar in Nerve Injury"

_jcm, 2021, doi:10.3390/jcm10194379_

Round 1

Reviewer 1 Report

Methods to detect nerve exposure are based on clinical inspection and it is sometime difficult depending the kind of socket you have. You can miss a lot on exposures when the roots have a particular anatomy.

It could have been interesting to have a post-op CBCT to confirm the results. In this sense, you exclude 193 cases with positive radiological contact and negative clinical exposure. Maybe some of these cases were positive with difficult sockets to explore.

Reviewer 2 Report

The present study reported on a significant clinical finding during the extraction of the lower wisdom tooth. The aim is clear, the data set relevant and the results clearly reported. It is suitable for publication within JCM after minor corrections:

  1. Introduction: please, add the following sentence at line 38. “Consequently, several surgical approaches have been proposed to reconstruct the damaged tissues, with promising short-term outcomes (Roccuzzo et al. 2021; https://doi.org/10.3390/app11093927).
  2. Discussion: please, add a paragraph with the limitations of the study (retrospective study, differences in n between the two groups etc.)

Round 2

Reviewer 1 Report

It is an observational study and the conclusions are related with their visual inspection. there are not methodological mistakes, but I are not sure if this is the most accurate methos to detect exposure. In other way, thea article is corect, well structured and written, but I am not sure if the content is clinically relevant
